# Elderly care expectation how to influence the fertility desire: Evidence from Chinese general social survey data

Jia Yang *

School of Public Administration, South China University of Technology, Guang Zhou,China

* payangjia@mail.scut.edu.cn

## Abstract

### Background

The changes of population structure will have profound impact on economic and social development. In recent years, the continuous intensification of population aging and the persistent decline of fertility rates have posed a certain challenge to the stability of population structure in China.This paper combines the two social hot issues of elderly care and fertility, and puts forward relevant policies and enlightenment through quantitative analysis and research,so as to promote better social development.

### Methods

Based on the data of the Chinese General Social Survey (CGSS) in 2017 and 2021, this paper uses the logistic regression model, linear regression model and the Propensity Score Matching model to analyze the influence of the elderly care expectation on residents' fertility desire. The elderly care expectations include government elderly care, self-care, offspring elderly care and responsibility sharing.The fertility desire includes 'whether to give birth', 'quantity','gender preference' and 'fertility echelon'.

### Result

The research results shows that: first, government elderly care expectations not only significantly reduces residents' willingness to have children, but also has an inhibitory effect on lower fertility echelons, but has a promoting effect on higher fertility echelons. Second, the self-care expectations significantly reduces residents' willingness to have children and has a significant negative impact on their expected fertility quantity, especially has a significant inhibitory effect on lower fertility echelons.Third, offspring elderly care expectations significantly increases residents' willingness to have children and has a significant positive impact on their expected fertility quantity, especially for increasing residents' willingness to have a second child, and residents with offspring elderly care expectations have a higher preference for boys.

**Data availability statement:** All data are available from the CGSS database. http://cgss.ruc.edu.cn/

**Funding:** The author(s) received no specific funding for this work.;

**Competing interests:** The authors have declared that no competing interests exist.

## Conclusion

This study highlights the complex relationship between elderly care expectation and fertility desire, offering valuable insights for policymakers. The concept of fertility echelons provides a nuanced understanding of the dynamic impacts on fertility preferences. By addressing both elderly care and fertility challenges, this research contributes to the development of strategies for sustainable social progress.

## 1. Introduction

The population situation affects economic development by influencing the effective supply of labor, the overall consumption level of society [1], and the level of government expenditure on public services. According to data released by the National Bureau of Statistics, China's natural population growth rate and birth rate had been at a low level from 2011 to 2020, with a continuous decrease in recent years, while the proportion of the population aged 65 and above had been steadily increasing.

In order to maintain China's advantage in human resources, implement strategies to actively address the aging population issue, and improve the country's population structure, on May 31, 2021, a meeting of the Political Bureau of the Central Committee of the Communist Party of China emphasized the implementation of a policy allowing couples to have up to three children along with supporting measures (https://www.peopleapp.com/column/30039199068-500004191137). It is evident that addressing the current issues is urgent. However, simply opening up birth policies is not sufficient; in-depth research on residents' fertility intentions is also necessary.

The decrease in the number of births has slowed down population growth momentum. Demographers utilized the "intention-behavior" model to assume that fertility behavior originates from fertility intentions. By analyzing fertility intentions, individual fertility behavior can be predicted, thereby forecasting the fertility level of the population [2]. Therefore, exploring the reasons for changes in population conditions necessitates an analysis of fertility intentions. Studies indicated that a fondness for children, suitable economic conditions, the desire for descendants, and support in old age were the primary reasons why young people were willing to have children [3]. Factors such as physical, psychological, political, economic, environmental, and educational considerations all impacted people's fertility intentions [4].

The research focus of the paper is on the influence of elderly care expectations on residents' fertility intentions, with elderly care expectations encompassing self elderly care, offspring elderly care, government elderly care, and shared elderly care responsibilities. Traditionally, fertility intentions mainly included three aspects: the desired number of children, gender preferences for children, and the timing of childbirth [5]. The fertility intentions examined in this paper primarily revolve around the decision to have children, the desired number of children, and gender preferences. The study on the desired number of children also encompasses high parity and low parity. By analyzing data from the Chinese General Social Survey (CGSS) in 2017 and 2021, the paper delved into the inherent connection between elderly care

expectations and residents' fertility, offering insights for the development and implementation of relevant policies and aiding decision-making to promote the coherent alignment between social fertility policies and economic policies [6], thereby facilitating the sustainable development of society and the economy [7].

## 2. Research background and hypotheses

In traditional Chinese fertility beliefs, there exist notions such as "lack of filial piety has three, no descendants is the greatest", "raising children to prevent old age", "passing on the family line", among others. However, with the development of society, the improvement of economic levels and social security systems, the role of the family has gradually weakened, and residents' fertility beliefs have undergone a transformation. Some residents consider child-rearing to require a significant amount of time, energy, and money, implying that the cost of child-rearing will be a considerable investment. Therefore, they view child-rearing as a rational investment for future old-age care, incorporating the "cost-benefit" theory into their reproductive decision-making, while considering both costs and expected benefits [8]. Scholars, in studying the relationship between child quality and quantity, view children as durable consumer goods. Therefore, parents must make reproductive decisions within the constraints of limited resources to maximize their own utility [9].

As Chinese traditional concept of elderly care responsibility evolves and changes, the subject of elderly care responsibility is gradually becoming more diversified, differentiated, and personalized [10]. Although traditional family elderly care concepts still dominate, they are weakening continuously, and forms of non-filial elderly care are gradually being accepted [11]. A sound social elderly care security system can partially replace the elderly care function of families, rationalize the distribution of elderly care concerns, and disperse expectations of elderly care onto children [12]. Therefore, hypotheses were proposed. H1a: government elderly care has a negative impact on residents' willingness to have children; H2a: The government elderly care expectation has a negative impact on the expected number of births.Women who hold the concept of "offspring elderly care" exhibit significantly higher willingness to have a second child compared to those with other elderly care beliefs [13]. As the number of children increases, elderly individuals receive relatively more economic support, life care, and emotional comfort [14,15], which may prompt residents to improve their own elderly care situations by having more children [16]. Therefore, hypotheses were proposed H1b: offspring elderly care expectation has a positive impact on residents' willingness to have children; H2b: offspring elderly care expectation has a positive impact on the expected number of births. Residents who choose "government elderly care" mainly seek basic livelihood security. However, government-led elderly care also faces issues such as limited elderly care resources, insufficient human resources, and low service professionalism [17]. Residents who choose "self-elderly-care" primarily seek efficiency and personalized elderly care needs [18]. Moreover, "self-elderly-care" has certain positive impacts, helping to reduce the burden of elderly care on society and families [19].Therefore, hypotheses were proposed. H1c: self elderly care expectation has a negative impact on residents' willingness to have children; H2c: self elderly care expectation has a negative impact on the expected number of births.

Studies indicate that elderly care expectations may have a certain impact on residents' willingness to have children. The emergence of basic elderly care insurance systems has partially replaced the elderly care responsibilities of some families and children, ensuring the basic living standards of the elderly while alleviating their psychological burdens. However, some traditional fertility beliefs still remain deeply rooted in relatively underdeveloped rural areas, where men are seen as the main providers of family productivity and primary bearers of elderly care responsibilities, leading to an increased desire to have male children. Elderly resources rely on personal savings and the number of children. If safeguards increase, the older generation may be less willing to bear higher social security taxes through younger generations, thereby reducing the number of descendants, which will decrease with the increase in social security taxes [20]. Studies on the relationship between elderly care insurance and residents' willingness to have children reveal significant impacts. Participation in elderly care insurance has a crowding-out effect on the number of children residents want to have. Furthermore, concerning the gender of children, residents who participate in elderly care insurance are more

inclined to have daughters. Participation in insurance can reduce one's own elderly care responsibilities and dependence on children in old age, thereby helping to alleviate concerns about elderly care [21,22].Research on the interactive relationship between social elderly care insurance, elderly care expectations, and willingness to have children shows that the willingness of residents who expect government elderly care and offspring elderly care to have children significantly decreases after they pay social elderly care insurance premiums. This indicates that elderly care responsibilities do have a certain impact on fertility intentions. Moreover, after paying social elderly care insurance premiums, those who expect government elderly care reduce the number of children more than those who expect offspring elderly care [23].

In the context of continuous social and economic development, residents' elderly care security systems are gradually improving, and elderly care choices are no longer limited to traditional methods of raising children for old age. Based on the above literature, it can be observed that elderly care expectations to some extent influence the reproductive situations of residents. Therefore, this article proposes the following research design.

## 3. Materials and Methods

### 3.1. Data source

The data used in this study is sourced from the China General Social Survey (CGSS) for the years 2017 and 2021. CGSS is a nationwide, comprehensive, and continuous large-scale social survey project with data that holds a certain level of authenticity and broad representativeness [24]. It aids in summarizing long-term social trends and provides data for government decision-making and international comparative research. In line with the research topic of this study, considering both residents' fertility intentions and their fertility capabilities, a sample of individuals aged 18–40 was selected as the research subjects. After removing missing and invalid samples, a total of 4722 valid samples were obtained.

### 3.2. Variable selection

**3.2.1. Variable definition.** Dependent Variable: The question from the CGSS survey questionnaire "If there were no policy restrictions, how many children would you like to have?" is defined as "desired number of births". Options of 3 children or more are categorized as 3. In defining the "Whether to give birth" variable, a value of "1" is assigned to those who wish to have children (choosing a number greater than 0) and "0" to those who do not wish to have children (choosing 0). In studying fertility gender preferences, the difference between the responses to "How many sons would you like to have?" and "How many daughters would you like to have?" is considered as the "gender preference" variable.

Independent Variable: The question from the CGSS survey about elderly care responsibility is "Who should primarily be responsible for the elderly with children?" This variable is named "elderly care expectation" and is set as the independent variable. Three dummy variables are introduced based on the answer options: "government/children/elders sharing responsibility" is used as the reference category to analyze the differences in residents' fertility intentions when considering "primarily responsible by children", "primarily responsible by the elderly themselves", and "primarily responsible by the government".

Control Variables: Apart from elderly care responsibilities, various factors influence fertility intentions, necessitating the inclusion of control variables for more accurate model estimation. Based on relevant literature, this study includes gender (female coded as 1, male coded as 2), age (calculated based on the respondent's birth year and survey time), ethnicity (Han ethnicity coded as 0, minority ethnicity coded as 1), family economic status (far below average coded as 1, below average coded as 2, average coded as 3, above average coded as 4, far above average coded as 5), health status (very unhealthy coded as 1, somewhat unhealthy coded as 2, average coded as 3, somewhat healthy coded as 4, very healthy coded as 5), basic pension insurance (not enrolled coded as 0, enrolled coded as 1), commercial pension insurance (not enrolled coded as 0, enrolled coded as 1), urban basic medical insurance/new rural cooperative medical care/public

medical care (not enrolled coded as 0, enrolled coded as 1), commercial medical insurance (not enrolled coded as 0, enrolled coded as 1), residence (urban coded as 1, rural coded as 2), survey time (2017 and 2021) as control variables added to the model.

### 3.2.2. Descriptive statistics of variables.

According to the comprehensive information in Table 1, among the valid samples, approximately 94.3% of individuals express a desire for children. About 20.2% of residents wish to have one child if there were no policy restrictions, while approximately 66.4% wish to have two children. the proportions of residents in the valid sample who expect their children, themselves, the government, and responsibility sharing for elderly care are 55.9%, 2.2%, 4.7%, and 37.2%, respectively. The data indicates that relying on offspring for elderly care is still the predominant expectation among residents. Although elderly care methods are continually evolving, the traditional position of relying on offspring for elderly care remains unchanged. According to data from the seventh national census released by the National Bureau of Statistics (http://www.stats.gov.cn/tjsj/zxfb/202105/t20210510_1817180.html), there are currently 34.9 million more males than females in China, with a sex ratio of 105.07 males per 100 females. Hence, it is necessary to control residents' fertility situations to adjust the population structure..

According to the comprehensive information in Table 2, when residents expect the government or themselves to be responsible for elderly care or when responsibility is shared, their demand for relying on children for elderly care decreases, leading to a lower fertility intention among residents. Additionally, when residents expect to rely on themselves for elderly care, the proportion of residents wanting to have children is the lowest, at 90.3%, indicating a significant substitution effect between self-care and relying on children for elderly care. However, when residents expect their children to be

**Table 1. Descriptive statistics.**

|  | Variable | Sign | Obs | Mean | SD |
|---|---|---|---|---|---|
| Dependent variable | Whether to give birth | *Fer* | *4722* | 0.943 | 0.231 |
|  | Expected number of births | *Bir* | *4722* | 1.761 | 0.670 |
|  | Gender preference | *BIG* | *4722* | −0.070 | 0.671 |
| Independent variable | Offspring elderly care | *CP* | *4722* | 0.559 | 0.497 |
|  | Self elderly care | *OP* | *4722* | 0.022 | 0.146 |
|  | Government elderly care | *GP* | *4722* | 0.047 | 0.212 |
|  | Sharing responsibility | *AP* | *4722* | 0.372 | 0.483 |
| Control variable | Gender | *Gen* | *4722* | 1.452 | 0.498 |
|  | Ethnicity | *Eth* | *4722* | 0.072 | 0.259 |
|  | Family economic status | *FEL* | *4722* | 2.716 | 0.681 |
|  | Health status | *Hea* | *4722* | 4.042 | 0.846 |
|  | Age | *Age* | *4722* | 30.168 | 6.294 |
|  | Social security | *Ins* | *4722* | 1.783 | 0.943 |
|  | Residence | *RR* | *4722* | 1.714 | 0.452 |
|  | Survey schedule | *Time* | *4722* | 2018.595 | 1.958 |

**Table 2. Distribution of fertility desire.**

| *Bir* | All samples | Elderly Caring Expectation | | | |
|---|---|---|---|---|---|
|  |  | Self | Offspring | Government | Share |
| 0 | 5.6% | 9.7% | 4.0% | 9.5% | 7.5% |
| 1 | 20.2% | 25.2% | 18.6% | 22.5% | 22.1% |
| 2 | 66.4% | 56.3% | 68.8% | 57.7% | 64.7% |
| 3 and over | 7.6% | 8.7% | 8.6% | 10.4% | 5.8% |

responsible for elderly care, the proportion of those willing to have children is the highest, reaching 96.0%. This suggests that residents who want to rely on their children for elderly care have a greater subjective dependence on their offspring, leading to an increased demand for children and consequently an increase in fertility intentions.It is noteworthy that when residents expect the government to be responsible for elderly care, the proportion of those wanting to have three or more children is the highest, at 10.4%.This indicates that government-supported elderly care may have a stimulating effect on higher-order fertility.

### 3.3. Model construction

In the model construction, the dependent variable "whether one wants to have children" is considered as a binary variable. The binary logistic regression is selected to analyze the impact of elderly care expectations on whether residents want to have children. The constructed model is as follows:

$$\ln\left(\frac{q_i}{1-q_i}\right) = \alpha + PR_i^T \cdot \beta + Con_i^T \cdot \chi + \delta_i \tag{1}$$

Considering the recent implementation of the "Two-Child Policy" and "Three-Child Policy" in the country, to further understand the impact of elderly care responsibility on residents' fertility intentions, the following model is constructed. Sub-sample regression analysis will be conducted as follows:

$$\ln\left(\frac{o_i}{1-o_i}\right) = \eta + PR_i^T \cdot \iota + Con_i^T \cdot \kappa + \lambda_i \tag{2}$$

Estimation of Resident Expectations on the Number of Children using Multiple Linear Regression Model. The constructed model is as follows:

$$Bir_i = \varepsilon + PR_i^T \cdot \phi + Con_i^T \cdot \gamma + \mu_i \tag{3}$$

In the study of residents' preferences for the gender of their children, the same multiple linear regression model as described above is utilized for analysis. The formula includes the following components:

   PR represents elderly care responsibility, comprising *CP,OP,GP,* and AP.Con denotes the control variables.For the formula,$i\in(1,2,...,n)$, where *n* is the total sample size. $q_i$ represents the probability of the occurrence of residents' expectations regarding childbirth. $o_i$ encompasses the probabilities of events where the expected number of children ranges from 1,1–2,2,2 to more,3 and above.$\alpha$、 $\varepsilon$、 $\eta$ indicate constant terms to be estimated.$\delta_i$、 $\lambda_i$、 $\mu_i$ represent random error terms.$\beta$、 $\chi$、 $\varphi$、 $\gamma$、 $\tau$、 $\kappa$ signify the vector of regression coefficients to be estimated.

## 4. Results

This study utilized Stata 15 software for data analysis. Before conducting regression analysis, a test for multicollinearity was performed using the variance inflation factor (VIF) on the sample data to ensure the accuracy of the regression coefficients. The test results indicated that the VIF values for all variables were below 1.2, indicating the absence of severe multicollinearity issues among the variables. Hence, the model passed the multicollinearity test.

### 4.1. Impact of elderly care expectations on residents' fertility intentions

**4.1.1. Impact of elderly care expectations on whether to give birth.** The likelihood ratio statistic LR chi-square value is significant, with Prob>chi2 = 0.000. Additionally, the estimated values of variables in each model are stable, indicating a good model fit and significant joint significance of the variables.

In Table 3, among the independent variables, the coefficient of CP is significantly positive at the 1% level. On the other hand, the coefficients of GP and OP are significantly negative at the 5% level. This suggests that expectations of offspring caring for parents have a significant positive impact on residents' desire to have children, while personal and governmental elderly care expectations have a significant negative influence. Specifically, choosing children for elderly care increases the probability of wanting to have children by 2.7%, while opting for personal or governmental elderly care decreases this probability by 4.1% and 2.7%, respectively.

Regarding control variables, the coefficient of Gen is significantly positive at the 10% level, indicating that males have a 1.2% higher probability of wanting to have children compared to females. The Age coefficient is significantly positive at the 1% level, suggesting that within a certain age range, residents' desire to have children increases with age. The Hea coefficient is significantly positive at the 1% level, indicating that as residents' health improves, their probability of wanting to have children also increases. The Ins coefficient is significantly negative at the 1% level, revealing that a higher level of social security leads to a lower probability of wanting to have children. Residents' purchase of old-age pension and medical insurance meets the basic living security of residents after they are old, and is conducive to preventing and resolving the risk of reproductive medical expenses, improving residents' confidence in fertility, and thus has a potential impact on fertility levels [25,26]. It also shows that there is a certain crowding-out effect of endowment insurance on fertility intention [27], and the medical insurance system also promotes people to choose non-child pension [28].The RR coefficient is significantly negative at the 1% level, indicating that rural residents have a 3.9% higher probability of wanting to have children compared to urban residents. The Time coefficient is significantly negative at the 1% level, suggesting that in 2021, the overall probability of residents wanting to have children is 1.3% lower than in 2017. All variable coefficients align with real-world scenarios, indicating a good model fit.

Table 3. The logistic regression results of elderly care expectation on whether to give birth.

| Variable | (1) | (2) | (3) | dy/dx |
|---|---|---|---|---|
| | Fer | Fer | Fer | |
| CP | 0.655*** (0.134) | / | 0.539*** (0.140) | 0.027*** |
| OP | −0.289 (0.345) | / | −0.819** (0.361) | −0.041** |
| GP | −0.260 (0.247) | / | −0.550** (0.262) | −0.027** |
| Gen | / | 0.279** (0.132) | 0.244* (0.133) | 0.012* |
| Age | / | 0.125*** (0.012) | 0.127*** (0.012) | 0.006*** |
| Eth | / | −0.219 (0.241) | −0.202 (0.244) | −0.010 |
| Hea | / | 0.216*** (0.079) | 0.213*** (0.079) | 0.011*** |
| FEL | / | 0.094 (0.099) | 0.076 (0.099) | 0.004 |
| Ins | / | −0.262*** (0.072) | −0.242*** (0.072) | −0.012*** |
| RR | / | −0.884*** (0.179) | −0.781*** (0.181) | −0.039*** |
| Time | / | −0.240*** (0.033) | −0.254*** (0.034) | −0.013*** |
| Constant | 2.519*** (0.091) | 483.790*** (67.216) | 511.375*** (68.366) | / |
| Prob > chi$^2$ | 0.000 | 0.000 | 0.000 | / |
| LR chi$^2$ | 32.19 | 220.28 | 251.10 | / |
| Pseudo R$^2$ | 0.016 | 0.107 | 0.122 | / |

Notes: standard error in parentheses,

*** $p < 0.01$,

** $p < 0.05$,

* $p < 0.1$.

**4.1.2. Impact of elderly care expectations on residents' expected number of births.** The regression results for residents' elderly care expectations on their expected number of children are presented in Table 4. The estimated values of variables in the model are stable, with Prob>F = 0.000, indicating a good model fit and significant joint significance of the variables.

The regression results indicate that compared to shared elderly care responsibilities, expectations of children caring for parents have a significant positive impact on residents' expected number of children at the 1% level. Conversely, expectations of self-care in old age have a significant negative impact on residents' expected number of children at the 10% level.

Regarding control variables, the coefficient of Gen is significantly positive at the 1% level, indicating that males have a higher expected number of children compared to females. The Age coefficient is significantly positive at the 1% level, suggesting that within a certain age range, residents' expected number of children gradually increases with age. The Eth coefficient is significantly positive at the 1% level, indicating that the expected number of children for ethnic minority residents is higher than for non-ethnic minority residents. The Hea coefficient is significantly positive at the 5% level, revealing that residents with higher health levels have a higher expected number of children. The Ins coefficient is significantly negative at the 10% level, suggesting that residents with higher social security levels have a lower expected number of children. The RR coefficient is significantly negative at the 1% level, indicating that rural residents have a higher expected number of children compared to urban residents. The Time coefficient is significantly negative at the 1% level, indicating that in recent years, the overall expected fertility level among residents may be gradually decreasing. All coefficients align with real-world scenarios, indicating a good model fit with a certain level of persuasiveness and explanatory power.

**4.1.3. Impact of elderly care expectations on residents' gender preference.** The regression results for residents' elderly care expectations on their gender preference for children are depicted in Table 5. The estimated

**Table 4. The regression results of elderly care expectation on the expected number of births.**

| Variable | (1) | (2) | (3) |
|---|---|---|---|
| | Bir | Bir | Bir |
| CP | 0.132*** (0.021) | / | 0.107*** (0.020) |
| OP | −0.047 (0.068) | / | −0.110* (0.066) |
| GP | 0.001 (0.047) | / | −0.038 (0.047) |
| Gen | / | 0.063*** (0.019) | 0.055*** (0.019) |
| Age | / | 0.016*** (0.002) | 0.016*** (0.002) |
| Eth | / | 0.156*** (0.037) | 0.153*** (0.037) |
| Hea | / | 0.026** (0.012) | 0.026** (0.011) |
| FEL | / | 0.014 (0.014) | 0.013 (0.014) |
| Ins | / | −0.024** (0.011) | −0.019* (0.011) |
| RR | / | −0.127*** (0.022) | −0.108*** (0.022) |
| Time | / | −0.034*** (0.005) | −0.036*** (0.005) |
| Constant | 1.688*** (0.016) | 69.426*** (9.886) | 72.923*** (9.891) |
| Prob >F | 0.000 | 0.000 | 0.000 |
| Adj R² | 0.009 | 0.044 | 0.051 |

Notes: standard error in parentheses,

*** p < 0.01,

** p < 0.05,

* p < 0.1.

 

values of variables in the model are stable, with Prob>F = 0.000, suggesting a good model fit and significant joint significance of the variables.

The regression results reveal that residents' elderly care expectations indeed have a significant impact on their gender preference for children. Specifically, the coefficient of CP is significantly positive at the 5% level, indicating that compared to shared elderly care responsibilities, expectations of children caring for parents lead residents to have a higher preference for having boys. This suggests that residents with expectations of children providing elderly care tend to place more emphasis on having sons, thereby increasing their preference for boys. This, to some extent, indicates that men in society may be shouldering more elderly care responsibilities.

The Gen coefficient is significantly positive at the 1% level, indicating that males have a higher preference for having boys. The Age coefficient is significantly positive at the 1% level, suggesting that within a certain age range, residents' preference for having boys gradually increases with age. The Ins coefficient is significantly negative at the 1% level, suggesting that residents with higher social security levels have a lower preference for having boys. The RR coefficient is significantly negative at the 1% level, indicating that rural residents have a higher expectation for having boys compared to urban residents.

These findings shed light on the intricate relationship between elderly care expectations and gender preferences for children among residents, highlighting how societal expectations and individual circumstances can influence family planning decisions and gender preferences.

**4.1.4. The impact of elderly care expectations on residents' fertility echelons.** Previous findings have confirmed that residents' elderly care expectations do indeed have an impact on whether they desire to have children and the expected number of children. However, the influence of residents' elderly care expectations on their fertility intentions may vary across different fertility parities. Therefore, in order to comprehensively explore the intrinsic relationship between the two, the article conducts sub-sample regressions to analyze the impact of elderly care expectations on residents' fertility parities.

**Table 5. The impact of elderly care expectation on gender preference.**

| Variable | (1) | (2) | (3) |
|---|---|---|---|
| | BIG | BIG | BIG |
| CP | 0.055*** (0.017) | / | 0.041** (0.017) |
| OP | −0.052 (0.055) | / | −0.072 (0.055) |
| GP | 0.021 (0.039) | / | 0.003 (0.039) |
| Gen | / | 0.057*** (0.016) | 0.054*** (0.016) |
| Age | / | 0.004*** (0.001) | 0.004*** (0.001) |
| Eth | / | 0.022 (0.031) | 0.021 (0.031) |
| Hea | / | 0.001 (0.010) | 0.001 (0.010) |
| FEL | / | 0.001 (0.012) | 0.001 (0.012) |
| Ins | / | −0.025*** (0.009) | −0.024*** (0.009) |
| RR | / | −0.046** (0.018) | −0.039** (0.018) |
| Time | / | −0.012*** (0.004) | −0.012*** (0.004) |
| Constant | 1.916*** (0.013) | 25.086*** (8.218) | 26.069*** (8.248) |
| Prob>F | 0.004 | 0.000 | 0.000 |
| F | 4.41 | 5.51 | 4.85 |
| $R^2$ | 0.003 | 0.009 | 0.011 |

Notes: standard error in parentheses,

*** p<0.01,

** p<0.05, * p<0.1.

According to the information in Table 6, the likelihood ratio test statistic LR chi$^2$ values for each model are relatively high, with a Prob>chi$^2$ value of 0.000, indicating good overall model fit and the joint significance of the variables. The model results indicate that when residents desire to have one child (compared to desiring zero children), the CP coefficient is significantly positive at the 5% level, while the OP and GP coefficients are significantly negative at the 5% and 10% levels, respectively. The expectation of offspring providing elderly care increases the probability of residents wanting to have one child by 5.1%, while self-provided elderly care and government-provided elderly care decrease this probability by 13% and 8%, respectively.

When residents desire to have two children (compared to desiring zero or one child), the CP coefficient is significantly positive at the 1% level, while the OP and GP coefficients are significantly negative at the 5% and 10% levels, respectively. The expectation of children providing elderly care increases the probability of residents desiring two children by 5.5%, while self-provided elderly care and government-provided elderly care decrease this probability by 8.9% and 6%, respectively.

For residents desiring three or more children (compared to desiring zero, one, or two children), the CP coefficient is significantly positive at the 1% level, and the GP coefficient is significantly positive at the 5% level. The expectation of children providing elderly care and government-provided elderly care increases the probability of residents desiring three or more children by 2.4% and 3.9%, respectively.

The regression results from Table 6 also indicate that the promotion effect of the expectation of children providing elderly care on residents' fertility intentions is highly significant. Government-provided elderly care expectations exhibit a

**Table 6. The influence of elderly care expectation on fertility echelons.**

| Variable | (1) | | (2) | | (3) | |
|---|---|---|---|---|---|---|
| | 0→1 | dy/dx | →2 | dy/dx | →3 and over | dy/dx |
| CP | 0.344** (0.160) | 0.051 | 0.281*** (0.073) | 0.055 | 0.345*** (0.127) | 0.024 |
| OP | −0.869** (0.419) | −0.130 | −0.453** (0.222) | −0.089 | 0.260 (0.368) | 0.018 |
| GP | −0.537* (0.306) | −0.080 | −0.305* (0.160) | −0.060 | 0.570** (0.247) | 0.039 |
| Gen | 0.109 (0.153) | 0.016 | 0.136** (0.069) | 0.027 | 0.235** (0.112) | 0.016 |
| Age | 0.120*** (0.014) | 0.018 | 0.031*** (0.006) | 0.006 | 0.051*** (0.010) | 0.004 |
| Eth | −0.712** (0.308) | −0.107 | 0.319** (0.150) | 0.062 | 0.946*** (0.162) | 0.065 |
| Hea | 0.220** (0.093) | 0.033 | 0.119*** (0.042) | 0.023 | −0.079 (0.064) | −0.005 |
| FEL | 0.188* (0.108) | 0.028 | −0.020 (0.052) | −0.004 | 0.168** (0.084) | 0.012 |
| Ins | −0.219*** (0.082) | −0.033 | −0.057 (0.038) | −0.011 | 0.018 (0.062) | 0.001 |
| RR | −0.727*** (0.203) | −0.109 | −0.292*** (0.082) | −0.057 | −0.278** (0.121) | −0.019 |
| Time | −0.191*** (0.039) | −0.029 | −0.100*** (0.017) | −0.020 | −0.039(0.029) | −0.003 |
| Constant | 384.175*** (77.979) | / | 201.501*** (35.263) | / | 75.233 (58.610) | / |
| Prob > chi$^2$ | 0.000 | / | 0.000 | / | 0.000 | / |
| LR chi$^2$ | 153.43 | / | 125.86 | / | 93.20 | / |
| Pseudo R$^2$ | 0.119 | / | 0.024 | / | 0.037 | / |

Notes: standard error in parentheses,

*** p<0.01,

** p<0.05,

* p<0.1. '0→1' indicates that when the expected number of births is 1 compared with the expected number of births is 0; '→2' indicates that when the expected number of births is 2, compared with the expected number of births of 0 and 1; '→3 and over' indicates that when the expected number of births is 3 or more, compared with the expected number of births of 0, 1 and 2.

significant inhibitory effect on residents at lower fertility parities and a significant promoting effect at higher fertility parities, showing a dynamic impact effect.

## 4.2. Robustness check

The regression results in the previous section indicate that residents' elderly care expectations do indeed have a certain impact on their fertility intentions. However, since the study utilizes non-experimental data, there may be data biases and significant confounding variable effects. To address this, propensity score matching (PSM) is employed to reduce systematic biases(k = 2,cal = 0.01). PSM is a statistical method used to reduce the impact of confounding variables in observational studies. The core idea is to simulate the effect of random experiments by estimating the propensity score of each individual, and then matching the individuals in the treatment group and the control group according to the propensity score.k represents the proportion of matching, k = 2 means that each treatment group matches two control groups. C represents the threshold for matching, c = 0.01 means that the difference between the search propensity score in the control group and the individual difference in the treatment group is less than 0.01.Both k and c determine the quality and quantity of matching. In the case of large sample size in the control group, increasing k and decreasing c appropriately can improve the matching accuracy.'Treated' represents the individual being treated. 'Controls' represent individuals that are not processed. 'Difference' represents the difference between the treatment group and the control group in the outcome variables.Subsequently, the analysis will reexamine the impact of residents' elderly care expectations on their fertility intentions after mitigating these biases.

Following propensity score matching, the standard deviations of most variables have significantly decreased. The impact of elderly care expectations on residents' fertility intentions will be reanalyzed post propensity score matching to account for potential biases. The changes in standard deviations before and after propensity score matching are illustrated in Fig 1.

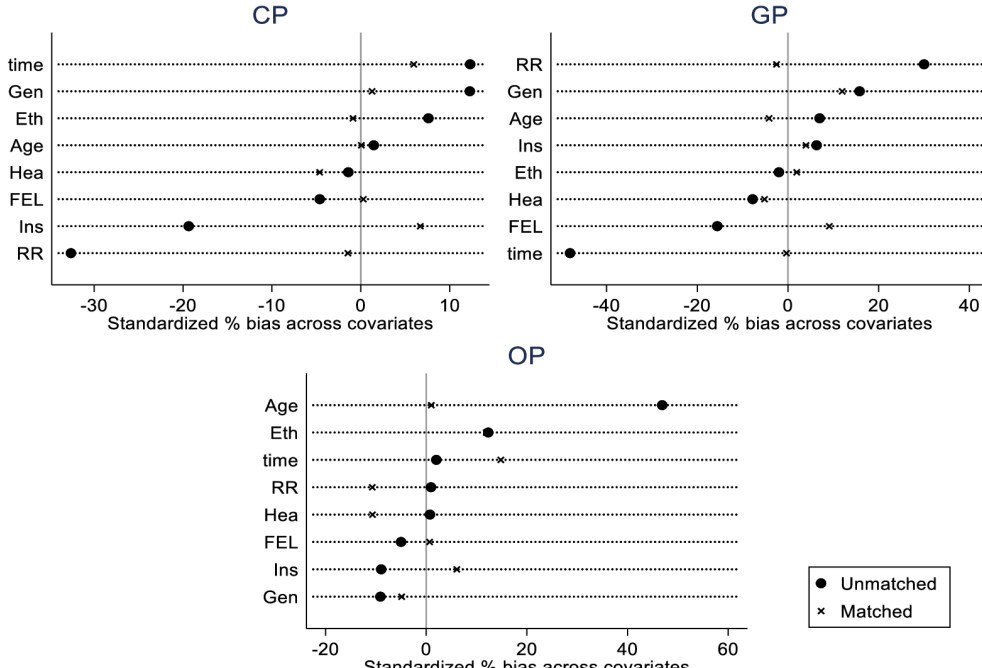

**Fig 1. Change chart of standard deviation before and after propensity score matching.t.**

According to the propensity score matching results in Table 7, compared to the equal sharing of elderly care responsibilities, the expectation of offspring elderly care has a significant positive impact on residents' desire to have children at a 1% level. Choosing to rely on children for elderly care increases the probability of residents wanting to have children by 3%. On the other hand, the expectation of self-provided elderly care has a significant negative impact on residents' desire to have children at a 5% level, with choosing self-provided elderly care resulting in a 6.5% decrease in the probability of residents wanting to have children. Similarly, the expectation of government-provided elderly care has a significant negative impact on residents' desire to have children at a 5% level, with opting for government-provided elderly care leading to a 5% reduction in the probability of residents wanting to have children. The consistency between the propensity score matching results and the logistic regression results confirms the validity of hypotheses H1a, H1b, and H1c. When residents perceive that children bear the main responsibility for elderly care, it naturally increases their desire to have children. Conversely, self-provided and government-provided elderly care relieve children of elderly care responsibilities, thereby reducing residents' reliance on their children and indirectly indicating a higher level of confidence in themselves and the government.

According to the propensity score matching results in Table 8, the expectation of children providing elderly care has a significant positive impact on residents' desired number of children at a 1% level. Choosing to rely on children for elderly care increases the overall desired number of children by approximately 9.8%. Studies indicate that an increase in the number of children also significantly enhances the elderly's elderly care capital, while simultaneously boosting financial and social capital but reducing material and human capital [29] Therefore, residents who opt for children to provide elderly care tend to have a higher desired number of children. On the other hand, the expectation of self-provided elderly care has a significant negative impact on residents' desired number of children at a 10% level. Choosing self-provided elderly care results in a decrease of 18.6% in the overall desired number of children. Residents who choose to rely on self-provided

**Table 7. Propensity score matching results of whether to give birth.**

|  |  | Treated | Controls | Difference (Bootstrap n=1000) | Z |
|---|---|---|---|---|---|
| ATT | CP | 0.960 | 0.936 | 0.024*** | 2.58 |
|  | OP | 0.903 | 0.984 | −0.081** | −2.43 |
|  | GP | 0.905 | 0.965 | −0.060** | −2.45. |
|  | AP | 0.925 | 0.954 | −0.029*** | −2.97 |

Notes:

*** p<0.01,

** p<0.05, * p<0.1. 'Treated' represents the individual that select CP, OP,GP or AP,and 'Controls' represent individuals that do not select CP, OP,GP or AP. 'Difference' represents the difference between the treatment group and the control group in 'whether to give birth'.

**Table 8. Propensity score matching results of expected number of births.**

|  |  | Treated | Controls | Difference (Bootstrap n=1000) | Z |
|---|---|---|---|---|---|
| ATT | CP | 1.819 | 1.728 | 0.092*** | 3.40 |
|  | OP | 1.641 | 1.834 | −0.193* | −1.94 |
|  | GP | 1.692 | 1.805 | −0.113* | −1.66 |
|  | AP | 1.688 | 1.784 | −0.096*** | −3.42 |

Notes:

*** p<0.01, ** p<0.05,

* p<0.1.'Treated' represents the individual that select CP, OP,GP or AP,and 'Controls' represent individuals that do not select CP, OP,GP or AP. 'Difference' represents the difference between the treatment group and the control group in 'expected number of births'.

Table 9. Propensity score matching results of fertility gender preference.

| | | Treated | Controls | Difference (Bootstrap n = 1000) | Z |
|---|---|---|---|---|---|
| ATT | CP | 1.970 | 1.924 | 0.046** | 2.09 |
| | OP | 1.864 | 1.878 | −0.014 | −0.17 |
| | GP | 1.941 | 1.969 | −0.027 | −0.50 |
| | AP | 1.916 | 1.974 | −0.058** | −2.55 |

Notes: *** p < 0.01,

** p < 0.05, * p < 0.1.'Treated' represents the individual that select CP, OP,GP or AP,and 'Controls' represent individuals that do not select CP, OP,GP or AP. 'Difference' represents the difference between the treatment group and the control group in 'fertility gender preference'.

elderly care generally have lower subjective fertility intentions [30].The consistency between the propensity score matching results and the logistic regression results confirms the validity of hypotheses H2a, H2b, and H2c. There still exists a certain correlation between elderly care preferences and the desired number of children among residents. The traditional beliefs of 'raising children for old age' and 'more children, more blessings' continue to influence people's desired number of children to a certain extent [31].

According to the propensity score matching results in Table 9, compared to the equal sharing of elderly care responsibilities, the expectation of children providing elderly care has a significant positive impact on residents' desire to have children at a 1% level. Choosing children for elderly care increases the probability of residents wanting to have children by 3%.Conversely, the expectation of self-provided elderly care has a significant negative impact on residents' desire to have children at a 5% level, with choosing self-provided elderly care leading to a 6.5% decrease in the probability of residents wanting to have children. Similarly, the expectation of government-provided elderly care has a significant negative impact on residents' desire to have children at a 5% level, resulting in a 5% reduction in the probability of residents wanting to have children when opting for government-provided elderly care.The propensity score matching results and logistic regression results both show that government elderly care has a negative impact on residents' willingness to have children, offspring elderly care expectation has a positive impact on residents' willingness to have children and self elderly care expectation has a negative impact on residents' willingness to have children.Therefore, the hypotheses(H1a, H1b and H1c)have been confirmed. When residents perceive that children bear the main responsibility for elderly care, it naturally increases their desire to have children. However, when residents expect to provide elderly care for themselves or rely on government-provided elderly care, which relieves children of elderly care responsibilities, it decreases the residents' need for children. This also indirectly indicates a higher level of confidence among residents in themselves and the government.

This study is based on logistic regression models, OLS regression models, and PSM models. Initially, the sample is analyzed using logistic regression to explore the impact of elderly care expectations on whether residents want to have children. Subsequently, multiple linear regression is employed to investigate the impact of elderly care expectations on residents' desired number of children and gender preferences for offspring. Finally, through sample regression analysis, the study examines the influence of elderly care expectations on the birth order of children.To test the robustness of the models, propensity score matching (PSM) is utilized for secondary analysis. The results from the PSM analysis are compared with the original models. The consistency in the signs of the estimated parameter coefficients and their significance indicates that the regression results obtained from the models chosen in this study exhibit a certain level of robustness.

## 5. Conclusion and Implications

The research findings demonstrate that residents' expectations of elderly care significantly influence their desire to give birth, desired number of births, gender preferences for offspring, and fertility echelons. Specifically, the expectations of government elderly care significantly reduce residents' willingness to have children, with an inhibitory effect on low fertility

echelons and a promoting effect on high fertility echelons, indicating a dynamic impact of governmental elderly care on fertility echelons. Due to the high cost of raising children, residents either opt for governmental elderly care and refrain from having children, or when their desired number of children reaches a certain level, residents gain confidence in continuing to have children by choosing governmental elderly care, reflecting the full trust residents have in governmental elderly care.The expectations of self elderly care significantly decrease residents' desire to have children, with a significant negative impact on the desired number of children, especially pronounced for low fertility echelons. Expectations of offspring providing elderly care significantly increase residents' desire to have children, with a significant positive impact on the desired number of children. Additionally, expectations of offspring providing elderly care also have a significant positive impact on residents' preference for having male offspring, demonstrating both confidence in their children and the irreplaceable role of family ties in traditional Chinese values.

There are some limitations in this study. Although it is confirmed that the expectation of elderly care has a significant impact on fertility desire, it is limited to China. As we all know, the population situation and cultural background of China are relatively special, and the Chinese people have the traditional ideas of ' raise children to provide against old age ' and ' carry on the family line ',so the research results may not be applicable to other countries.But the latest research all over the world which related to this study through empirical research and sorting out relevant literature also finds that the increase in caregiving responsibilities toward elderly parents will lead to an immediate and substantial decline in the fertility expectations of adult children [32].

With the rapid economic development leading to increasing childcare costs and a continuous decline in fertility intentions becoming a prevalent phenomenon, the insights from the above research suggest that certain measures can be taken to adjust public fertility intentions:

Optimize fertility policies: Provide fertility and childcare subsidies in different tiers under the premise of loosening fertility restrictions; offer preferential checks and care services for pregnant women and infants to ensure their physical and mental health; strictly enforce maternity leave policies to ensure the wages, bonuses, and welfare benefits of parents during maternity leave and caregiving leave.

Enhance medical insurance coverage: Innovate medical insurance services, expand the depth of medical insurance coverage, improve maternity medical insurance coverage to boost residents' confidence in childbirth; encourage active participation in medical insurance, provide discounts based on birth order to broaden the coverage of medical insurance; offer flexible maternity insurance schemes for different demographics to effectively increase fertility intentions among various groups.

Promote Rational Allocation of Medical, Education, and Housing Resources: Strictly regulate the construction and management of medical facilities, educational institutions, and affordable housing, particularly focusing on rural areas. Through regional balanced development, pressure in urban areas can be alleviated, meeting residents' basic needs for medical care, education, and housing when having multiple children. Establish high-quality medical and educational teams, recruit skilled personnel with work experience and good ethics, and train a competent, ethical, law-abiding, meticulous workforce to ensure the continuous delivery of high-quality medical and educational services.

Enhance Government Elderly Care Support: Coordinate the use of community elderly care funds, optimize the structure of fiscal spending on elderly care [33], encourage broad participation from social entities in the construction of community elderly care facilities to provide services such as living, dining, medical care, and emotional support for the elderly. Define standards for managing the elderly care service industry, strengthen workforce development in elderly care services to enhance the capacity of caring for the elderly, effectively safeguarding their living standards and health. Establish a sound system for supporting families with special circumstances in family planning, improve the mechanism for government-led care involving social organizations to reduce the economic and psychological burdens on residents during childbirth.

Strengthen Contemporary Residents' Family Values: Chinese culture places strong emphasis on family values, prioritizing filial piety and the continuation of the family line. However, with the fading of these values among young people, activities like "Family Activity Weeks" organized by communities and workplaces can be promoted. Encourage young individuals to volunteer at community nursing homes, support schools in offering parent-child education programs to enhance communication and connections among family members, creating a harmonious and warm family environment. This can deepen residents' family values, thereby boosting fertility intentions.

These measures aim to address the declining fertility rates by improving support systems, optimizing resource allocation, enhancing healthcare and education services, strengthening elderly care, and reinforcing traditional family values within the evolving societal landscape.

## Author contributions

**Conceptualization:** Jia Yang.

**Data curation:** Jia Yang.

**Formal analysis:** Jia Yang.

**Funding acquisition:** Jia Yang.

**Investigation:** Jia Yang.

**Methodology:** Jia Yang.

**Project administration:** Jia Yang.

**Resources:** Jia Yang.

**Software:** Jia Yang.

**Supervision:** Jia Yang.

**Validation:** Jia Yang.

**Visualization:** Jia Yang.

**Writing – original draft:** Jia Yang.

**Writing – review & editing:** Jia Yang.

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
