## [Decision Letter · Decision Letter 0]

11 Feb 2025

PONE-D-24-51386Elderly Care Expectation how to Influence the Fertility Desire: Evidence from Chinese General Social Survey DataPLOS ONE

Dear Dr. Yang,

Thank you for submitting your manuscript to PLOS ONE. After careful consideration, we feel that it has merit but does not fully meet PLOS ONE’s publication criteria as it currently stands. Therefore, we invite you to submit a revised version of the manuscript that addresses the points raised during the review process.

The manuscript has been evaluated by two reviewers, and their comments are available below.

The reviewers have raised a number of major concerns. Could you please carefully revise the manuscript to address all comments raised?

We look forward to receiving your revised manuscript.

Kind regards,

Steve Zimmerman, PhD

Senior Editor, PLOS One

2. Please note that your Data Availability Statement is currently missing a direct link to access each database. If your manuscript is accepted for publication, you will be asked to provide these details on a very short timeline. We therefore suggest that you provide this information now, though we will not hold up the peer review process if you are unable.

Reviewers' comments:

Reviewer's Responses to Questions

**Comments to the Author**

1. Is the manuscript technically sound, and do the data support the conclusions?

Reviewer #1: Yes

Reviewer #2: Partly

2. Has the statistical analysis been performed appropriately and rigorously? 

Reviewer #1: I Don't Know

Reviewer #2: No

3. Have the authors made all data underlying the findings in their manuscript fully available?

Reviewer #1: Yes

Reviewer #2: Yes

4. Is the manuscript presented in an intelligible fashion and written in standard English?

Reviewer #1: Yes

Reviewer #2: Yes

5. Review Comments to the Author

Reviewer #1: This submission does not adhere to the conventional structure expected of academic papers, resembling more of a report or thesis, which may render it tedious for readers. The introduction is excessively lengthy, which detracts from the overall flow of the paper. Additionally, the article lacks a suitable structured abstract.

Furthermore, all references are predominantly related to the context of the study and largely focus on China. The results has not been adequately analyzed nor compared with studies from other countries, making it difficult to ascertain the broader implications of the findings. Notably, the results and discussion sections are not clearly distinguishable from one another.

The number of tables included is excessive, which may overwhelm the reader. Moreover, the limitations of the study are not clearly outlined, raising concerns about the study's rigor.

he generalizability of the findings is also a concern; given the study's limited context, it is unclear how applicable the results are to other settings or populations. A discussion on the potential applicability of the results in different cultural or geographical contexts would enhance the paper's contribution to the literature.

Before any further evaluation, the article should be revised to conform to academic standards. It should be prepared in a concise manner for readers, including fundamental components of an academic paper and following the formatting guidelines of the journal.

Nonetheless, the topic of the study is indeed interesting and holds potential value.

Reviewer #2: According to the authors they attempted to combine the two social issues of elderly care and fertility, and put forward what they describe as relevant policies and enlightenment through quantitative analysis and research, so as to promote better social development. The paper is mostly descriptive.

They did much work and the paper is generally well written. However, it is cumbersome to read and keeping track of all the abbreviations was a challenge. The modeling made sense. However, the manuscript lacks clarity in several palaces.

The propensity scoring is not well explained or presented. What was the cutoff threshold to declare something well matched? Although the sample size is quite adequate how was the sample size effected after all the matching? Table 8 is poorly labeled. Tell the reader what the treatment and control is (presumably the empirical sample vs. the propensity sample. If not, then what exactly?). Footnotes would be helpful. They make the statement, “To address this, propensity score matching (PSM) is employed to reduce systematic biases(k=2,cal=0.01)” What are k and cal? A statistician would perhaps understand, but some readers may need help with this. Also, why make the statement that, “The consistency between the propensity score matching results and the logistic regression results confirms the validity of hypotheses H1a, H1b, and H1c.”? Table 8 shows some rather large Z’s or is the reviewer missing something. Please clarify?

Also the interpretations of the findings are their interpretations. Was any attempt made to get some expert qualitative interpretation of the findings from the survey? In addition, it seems that the tables in the supplement are redundant. Please edit the paper for statistical clarification as noted and in general. Also, please give an objective assessment of any other limitations.

6. PLOS authors have the option to publish the peer review history of their article (what does this mean? ). If published, this will include your full peer review and any attached files.

**Do you want your identity to be public for this peer review?** For information about this choice, including consent withdrawal, please see our Privacy Policy .

Reviewer #1: No

Reviewer #2: No

---

## [Author Response · Author response to Decision Letter 1]

26 Feb 2025

First, I greatly appreciate your comment regarding the appropriateness and rigor of my statistical analysis.In this study, I employed logistic regression model, linear regression model and the Propensity Score Matching model, which are widely recognized and recommended in the field of social statistics for analyzing data of this nature. I selected these methods because they are well-suited to handle our type of data and effectively address our research questions. Specifically, "Whether to give birth" is a binary variable, so it is suitable for logistic regression model analysis. For the “expected number of births” and “gender preference”, they are more suitable for the least squares regression model to analyze.The non-experimental data used in the study may have data bias and the influence of confounding variables is more, so the propensity score matching model can be used to further verify the analysis.The role of PSM is to reduce confounding bias(reduces the impact of these confounding variables by balancing the distribution of covariates between groups, so as to estimate the treatment effect more accurately), simulate randomized trials(simulates the conditions of randomization experiments by matching individuals with similar propensity scores,thereby approximating the causal effect), and improve inter-group comparability(it ensures that the two groups are comparable in baseline characteristics by matching individuals with similar characteristics in the treatment group and the control group, thereby improving the reliability of the results).

To ensure the rigor of our analysis, I conducted several steps during the data preprocessing phase, including outlier handling, missing data imputation etc, and adhered strictly to relevant statistical guidelines and protocols.The results have shown consistency across multiple repeated experiments, further validating the reliability of our methods.

Second,I greatly appreciate your comment regarding the need for a more structured abstract. I agree that a well-organized abstract can significantly improve the clarity and accessibility of the article.In response to your suggestion, I have revised the abstract to follow a structured format,including the following sections:Background,Methods,Results, and Conclusions. This new structure provides a clearer and more comprehensive overview of our study, making it easier for readers to understand the key aspects of our research.

Third,You mention the format of the paper and the long citation. I have modified some of the formats and appropriately reduced the content of the citation to meet academic standards.

Fourth,you also mention the references, the adaptability of the research results and the limitations of the research.We acknowledge that the initial version lacked sufficient references to studies from other countries.Although it is confirmed that the expectation of elderly care has a significant impact on fertility desire, it is limited to China. As we all know, the population situation and cultural background of China are relatively special, and the Chinese people have the traditional ideas of ' raise children to provide against old age ' and ' carry on the family line ',so the research results may not be applicable to other countries.In this study, I added the latest international research literature related to this study, and the results found that the increase in caregiving responsibilities toward elderly parents will lead to an immediate and substantial decline in the fertility expectations of adult children, which are similar to our study.So it can show that the results of this study have certain applicability and provide a broader context for our research.

Fifth,about the problem of ‘lacks clarity’, I have been modified in the article.I have added appropriate annotations below the table and added appropriate interpretation of psm in the article to facilitate the reader's understanding, including the interpretation of ' Treated ', ' Controls ' and ' Difference ' in the table, and the interpretation of C and K in psm. After all the matching, more than 95 % of the samples are matched, indicating that the matching effect is good. However, due to the existence of different elderly care expectations and the existence of three outcome variables, the number of matching is more, so the effect of each sample matching is not described in detail in the article.

Sixth,We acknowledge that the original manuscript was somewhat cumbersome to read due to excessive use of abbreviations. In the revised version,we have tried to simplify the language and reduced unnecessary descriptive content to improve readability.

Seventh,We have expanded the discussion of the statistical results to address your concerns:explained why the consistency between the PSM results and logistic regression results supports hypotheses H1a, H1b, and H1c . The results of logistic regression show that expectations of offspring caring for parents have a significant positive impact on residents' desire to have children, while personal and governmental elderly care expectations have a significant negative influence. According to the results in Table 7, when cp, op and gp were selected, the values of difference were 0.024, -0.081 and-0.060, respectively, indicating that the average fertility intention of the treatment group was higher than that of the control group when cp was selected. When op was selected, the average fertility intention of the treatment group was lower than that of the control group. When gp was selected, the average fertility intention of the treatment group was lower than that of the control group.The results of the two studies were the same, which confirm H1a,H1b,H1c.

Eighth,i want to clarify the significance of the large Z-values in Table 8 here.The greater the z value, the stronger the significance of the difference between the treated group and the control group.The z value of cp is greater than that of op and gp, indicating that when cp is selected, the difference between the treated group and the control group is more significant, which is the same as the regression results in Table 4, indicating that the results are reasonable.

---

## [Decision Letter · Decision Letter 1]

26 Mar 2025

Elderly Care Expectation how to Influence the Fertility Desire: Evidence from Chinese General Social Survey Data

PONE-D-24-51386R1

Dear Dr. Yang,

We’re pleased to inform you that your manuscript has been judged scientifically suitable for publication and will be formally accepted for publication once it meets all outstanding technical requirements.

Kind regards,

Chhabi Lal Ranabhat

Academic Editor

PLOS ONE

Additional Editor Comments (optional):

Reviewers' comments:

Reviewer's Responses to Questions

**Comments to the Author**

1. If the authors have adequately addressed your comments raised in a previous round of review and you feel that this manuscript is now acceptable for publication, you may indicate that here to bypass the “Comments to the Author” section, enter your conflict of interest statement in the “Confidential to Editor” section, and submit your "Accept" recommendation.

Reviewer #1: All comments have been addressed

Reviewer #2: All comments have been addressed

2. Is the manuscript technically sound, and do the data support the conclusions?

Reviewer #1: Yes

Reviewer #2: (No Response)

3. Has the statistical analysis been performed appropriately and rigorously? 

Reviewer #1: Yes

Reviewer #2: (No Response)

4. Have the authors made all data underlying the findings in their manuscript fully available?

Reviewer #1: Yes

Reviewer #2: (No Response)

5. Is the manuscript presented in an intelligible fashion and written in standard English?

Reviewer #1: Yes

Reviewer #2: (No Response)

6. Review Comments to the Author

Reviewer #1: I think that the authors did a great job addressing reviewers' comments. The punctuation is incorrect in some instances.

no more comments.

Reviewer #2: (No Response)

7. PLOS authors have the option to publish the peer review history of their article (what does this mean? ). If published, this will include your full peer review and any attached files.

**Do you want your identity to be public for this peer review?** For information about this choice, including consent withdrawal, please see our Privacy Policy .

Reviewer #1: No

Reviewer #2: No

---

## [Editor Report · Acceptance letter]

PONE-D-24-51386R1

PLOS ONE

Dear Dr. Yang,

I'm pleased to inform you that your manuscript has been deemed suitable for publication in PLOS ONE. Congratulations! Your manuscript is now being handed over to our production team.

Kind regards,

on behalf of

Dr. Chhabi Lal Ranabhat

Academic Editor

PLOS ONE